# Synthesis, Structure, and Characterizations of a Volatile/Soluble Heterometallic Hexanuclear Precursor [NaMn_2_(thd)_4_(OAc)]_2_

**DOI:** 10.3390/molecules28237795

**Published:** 2023-11-27

**Authors:** Yuxuan Zhang, Zheng Wei, Evgeny V. Dikarev

**Affiliations:** Department of Chemistry, University at Albany, State University of New York, 1220 Washington Avenue, ETEC, Albany, NY 12226, USA; yzhang50@albany.edu (Y.Z.); zwei@albany.edu (Z.W.)

**Keywords:** single-source precursor, heterometallic complex, hexanuclear molecule, sodium-manganese oxide cathode materials

## Abstract

The paper describes a hetero*bi*metallic mixed-ligand hexanuclear precursor [NaMn_2_(thd)_4_(OAc)]_2_ (**1**) (thd = 2,2,6,6-tetramethyl-3,5-heptadionate; OAc = acetate) that was designed based on its lithium homoleptic analogue, [LiMn_2_(thd)_5_], by replacing one of the thd ligands with an acetate group in order to accommodate 5-coordinated sodium instead of tetrahedral lithium ion. The complex, which is highly volatile and soluble in a variety of common solvents, has been synthesized by both the solid-state and solution methods. The unique “dimer-of-trimers” heterometallic structure consists of two trinuclear [NaMn^II^_2_(thd)_4_]^+^ units firmly bridged by two acetate ligands. X-ray diffraction techniques, DART mass spectrometry, ICP-OES analysis, and IR spectroscopy have been employed to confirm the structure and composition of the hexanuclear complex. Similar to the Li counterpart forming LiMn_2_O_4_ spinel material upon thermal decomposition, the title Na:Mn = 1:2 compound was utilized as the first single-source precursor for the low-temperature preparation of Na_4_Mn_9_O_18_ tunnel oxide. Importantly, four Mn sites in the hexanuclear molecule can be potentially partially substituted by other transition metals, leading to hetero*tri*- and *tetra*metallic precursors for the advanced quaternary and quinary Na-ion oxide cathode materials.

## 1. Introduction

The first single-source molecular precursor for the lithium-manganese spinel cathode material, [LiMn_2_(thd)_5_] (thd = 2,2,6,6-tetramethyl-3,5-heptadionate), was reported several years ago [1]. This hetero*bi*metallic structure was a result of a long, systematic search for appropriate ligands and was initially regarded as a unique trinuclear architecture accommodating Mn^II^ ions only [1]. However, subsequent research [2,3] has revealed that one of two Mn sites can be partially or fully substituted by Co^II^, thus producing hetero*tri*metallic complexes [LiMn_2−x_Co_x_(thd)_5_] (0 < x ≤ 1) that initiated single-source precursors for the so-called 5V spinel cathodes [4,5,6]. This modification from hetero*bi*metallic to hetero*tri*metallic assembly was found not only to retain the structure and metal ratio but also to preserve the most vital thermodynamic characteristics of precursors, such as their solubility, volatility, sensitivity, and low-temperature decomposition [1,2,3].

It seems logical to use the model [LiMn_2_(thd)_5_] structure to design the first heterometallic complex with a Na:Mn ratio of 1:2 by simply substituting Na for Li. This would effectively create a single-source precursor for sodium-manganese oxide (SMO) cathodes. Among those, tunnel oxide, Na_4_Mn_9_O_18_ (Na_0.44_MnO_2_), is certainly the most attractive target due to its unique crystal structure, containing large-size tunnels for fast sodium de/insertion [7,8,9,10]. Layered oxides, P2-Na_x_MnO_2_ (x ≤ 0.67), are also widely studied, though those usually require second transition metal doping [11,12] to improve the electrochemical characteristics. Calcium-ferrite type NaMn_2_O_4_, which can potentially perform as a cathode material, has been synthesized at high-temperature and high-pressure conditions only [13,14].

It should be noted that no single-source precursor has been advanced for the preparation of the above-mentioned SMO cathode materials. While we have recently introduced [15] a broad family of precursors for layered P2-Na_x_MO_2_ oxides, the latter did not include Mn compounds. That said, the multi-source precursor approach has been successfully applied [16,17,18] for the hydrothermal preparation of Na_4_Mn_9_O_18_, though the product was found to be not a phase-pure.

Replacing Na for Li seems to be a daunting task compared to the substitution of one 3D transition metal with another due to the difference in coordination numbers of alkali metals. While a Li ion commonly appears as tetrahedrally coordinated in heterometallic complexes with transition metals [1,2,3,19,20,21], Na prefers higher coordination numbers. It typically experiences a coordination number of five or six upon participation in chelating bridging frameworks [22,23,24,25,26,27,28,29,30]. Great illustrations of this statement are analogous tetranuclear complexes with an A:M = 1:1 ratio, [A_2_M_2_L_6_] (A = Li, Na; M = transition metal) [27,31], where Na not only requires an additional coordination by THF molecules but also changes the whole ligand distribution within the heterometallic assembly. Even more, in none of the heterometallic structures [1,15,27,32] were we able to replace Na for Li, and vice versa, without changing the structure itself, as well as the connectivity pattern. The sole exception is the family of 1D polymeric chain complexes [AML_3_]_∞_ (L = acac = acetylacetonate or hfac = hexafluoroacetylacetonate) that accommodate both Li [31] and Na [33] as 6-coordinated “naked” ions. It is worth mentioning that the latter structure type also supports the K ion [34].

In this paper, we successfully utilized the model [LiMn_2_(thd)_5_] structure in order to design a heterometallic precursor with a Na:Mn = 1:2 ratio by replacing one of the thd ligands with an acetate group that assists in satisfying the Na ion coordination requirements. The [NaMn_2_(thd)_4_(OAc)]_2_ (**1**) complex possesses a unique hexanuclear “dimer of trimers” structure, features good volatility and solubility as its parent Li analog [1], and was confirmed to act as the first single-source precursor for the tunnel Na_4_Mn_9_O_18_ oxide.

## 2. Results and Discussion

### 2.1. Design of the Heterometallic Precursor

The trinuclear heterometallic molecule [LiMn_2_(thd)_5_] (Figure 1a) contains two Mn^II^ centers in distorted octahedral geometry and one tetrahedrally coordinated Li ion. Among the five thd ligands, two chelate Mn centers, the third group chelates Li/bridges to Mn, and the remaining two ligands act in an unusual manner of *tris*-bridging the metal atoms [1]. For replacing the tetrahedral Li by Na ion with a higher coordination number, at least one extra coordination should be provided, as has been accomplished in [A_2_M_2_L_6_] complexes by adding THF molecules [26]. However, it has been found that the insertion of even a small monodentate neutral ligand effectively destroys this very strained structure built on bulky thd ligands acting in very unusual coordination modes. Therefore, we shifted to replacing one of the thd ligands with a small monoanionic group that can provide even more efficient bridging, while opening up some space around the alkali metal center. The acetate group perfectly fits the bill (Figure 1b) and more, thus resulting in “dimerization” of the [LiMn_2_(thd)_4_(OAc)] trinuclear units (Figure 1c).

### 2.2. Synthesis and Properties of Heterometallic Precursor

[NaMn_2_(thd)_4_(OAc)]_2_ (**1**) has been obtained using both the solid-state and solution methods with the stoichiometric reaction (**1**). Block-shaped yellow crystals grown using the solid-state technique were initially checked by the ICP-OES analysis to reveal the ratio of Na:Mn as 1:2. Single crystals of **1** are extremely sensitive to oxygen, immediately changing to a brown color when exposed to air. The complex is soluble in most common organic solvents such as hexanes, dichloromethane, acetone, and alcohols. Strongly coordinating solvents like DMSO and H_2_O appeared to break up the heterometallic structure into homometallic fragments. Similar to its lithium counterpart, the hexanuclear complex exhibits good volatility, with the sublimation starting at ca. 100 °C under the static vacuum in a sealed ampule. Upon continuous heating, the yellow crystals of **1** appear to decompose at around 180 °C, accompanied by a sharp color change from yellow to brown.
8Na(thd) + 4Mn(OAc)_2_ → [NaMn_2_(thd)_4_(OAc)]_2_ + 6Na(OAc)(1)

The synthesis of [NaMn_2_(thd)_4_(OAc)]_2_ (**1**) can be scaled up using the solution method. Remarkably, the yellow bulk product obtained from the solution is a bit less sensitive when exposed to moist air than the one obtained by the solid-state approach. Powder X-ray diffraction was recorded for checking the purity of the bulk material prepared from the solution reaction (Figure 2). The Le Bail fit confirms that the experimental powder pattern of the bulk product corresponds well to the theoretical spectrum calculated from the single-crystal X-ray data (Figure 2 and Table 1).

### 2.3. Molecular Structure of Hexanuclear Precursor ***1***

A single-crystal X-ray analysis revealed that the hetero*bi*metallic precursor **1** consists of discrete hexanuclear molecules [NaMn_2_(thd)_4_(OAc)]_2_ (Figure 3) with a Na:Mn ratio of 1:2. It crystallizes in the centrosymmetric triclinic unit cell with the inversion center located in the middle of the assembly. There are two crystallographically independent “half-molecules” [NaMn_2_(thd)_4_(OAc)] in the unit cell, with the only significant difference between those being the degree of disorder of *tert*-butyl groups in thd ligands. Similar to its parent lithium analog, all three metal centers in **1** are chiral; however, the whole molecule is *meso* by virtue of two enantiomeric parts.

The [Mn_2_(thd)_4_] part in **1** is quite similar to the corresponding fragment in the parent [LiMn_2_(thd)_5_] structure, judging by the Mn–O bond lengths (Table 2) and O–Mn–O angles. Both Mn ions maintain a distorted octahedral geometry [35]. Four thd ligands act exactly in the same manner, as was observed before [1]: two groups purely chelate to Mn centers, while the other two ligands bridge both Mn and Na ions.

The major difference between the two structures is in the shape of the Mn–A–Mn (A = Li, Na) triangles, with elongation of the Mn–Na distances and sharpening of the Mn1–Na–Mn2 angle in **1** to accommodate for the higher coordination number of the Na ion (Table 2).

The coordination of the sodium ion by two oxygen atoms from diketonate groups and three oxygen atoms from acetate ligands can be described as either distorted square pyramidal or distorted vacant octahedral, according to calculations by the Continuous Shape Measures (CShM) method [36,37]. An average Na–O bond distance of 2.367 (3) Å in **1** (Table 2) is typical for the 5-coordinated sodium ion in a diketonate/acetate coordination environment [22,23,24,25]. Each of the two *μ*_3_-acetates are chelating Na and bridging it with Mn on one end and with another Na on the other end. These carboxylate groups are primarily responsible for bringing two [NaMn_2_(thd)_4_] units together in order to form a unique “dimer of trimers” structure.

### 2.4. Direct Analysis of Real Time (DART) Mass Spectrometry of Complex ***1***

Direct Analysis in Real Time (DART) mass spectrometry was performed to assess the retention of the heterometallic structure in the gas phase. In the positive mode (Figure 4), the mass spectrum features the [Na_2_Mn_4_(thd)_7_(OAc)_2_]^+^ ([M-thd)]^+^) peak (meas/calcd = 1665.738/1665.728), which confirms the retention of the hexanuclear structure. The high intensity of the [NaMn_2_(thd)_4_]^+^ peak (meas/calcd = 865.424/865.420) indicates the breaking of the molecule of **1** in half upon the loss of acetate ligands. All heterometallic ion fragments are shown in Table 3. Importantly, the analysis of all fragment peaks unambiguously confirms the (+2) oxidation state of the Mn ions.

### 2.5. Thermal Decomposition of Heterometallic Precursor

A thermal gravimetric analysis (TGA) of [NaMn_2_(thd)_4_(OAc)]_2_ (**1**) (Figure 5) was carried out by heating up precursor powder at 1 °C/min heating rate with 25 mL/min argon protection flow. Complex **1** shows a mass loss due to volatility at around 130 °C, followed by a sharp mass loss between 160 and 300 °C corresponding to the decomposition of the precursor and release of volatiles such as H_2_O, CO, CO_2_, and C_x_H_y_ from the ligands. The weight decrease at higher temperatures can be attributed to the loss of Na [27].

From the TGA curve, at 750 °C the total weight loss is 88.2%. The theoretical mass loss is 89.5%, calculated by using tunnel oxide Na_0.44_MnO_2_ as the decomposition product residue. The thermal decomposition of [NaMn_2_(thd)_4_(OAc)]_2_ (**1**) was carried out by a two-step method. The bulk microcrystalline powder of a heterometallic precursor was first preheated at 200 °C under argon flow in an atmosphere furnace for 30 min. The temperature was selected based on the TGA investigation (Figure 5) to get rid of most of the volatile components. After that, the residue was placed into a crucible and decomposed in a non-gradient furnace at 700 °C for 16 h in open air. The decomposition product was then investigated by X-ray powder diffraction to reveal a pure phase of Na_4_Mn_9_O_18_ oxide (sometimes designated as Na_0.44_MnO_2_). The Le Bail fit was performed (Figure 6) to confirm that the experimental powder pattern of the decomposition product corresponds to the data in the PDF-2 database (Table 4). The ICP-OES analysis of the decomposition product showed the ratio of Na:Mn as 0.46:1, corresponding well to the target material and also confirming a slight sodium loss from the hexanuclear precursor (Na:Mn = 1:2) due to the temperature used for pyrolysis.

## 3. Materials and Methods

### 3.1. Materials and Measurements

Manganese(II) acetate [Mn(OAc)_2_] and sodium methoxide [NaOCH_3_] were purchased from Sigma-Aldrich and used as received after checking their X-ray powder diffraction patterns. 2,2,6,6-Tetramethyl-3,5-heptadione (Hthd) was purchased from Sigma-Aldrich (Burlington, MA, USA) and used as received after checking its ^1^H NMR spectrum. The ICP-OES analysis was carried out on ICPE-9820 plasma atomic emission spectrometer, Shimadzu (Kyoto, Japan). The DART-MS spectra were recorded on AccuTof 4G LC-plus DART mass spectrometer, JEOL (Tokyo, Japan). The IR spectrum was measured using IRTracer-100 Fourier Transform Infrared Spectrophotometer, Shimadzu. X-ray powder diffraction data were collected on a Rigaku multipurpose *θ*-*θ* X-ray SmartLab SE diffractometer (Rigaku, Tokyo, Japan) (Cu K*α* radiation, HyPix-400 two-dimensional advanced photon counting hybrid pixel array detector (Rigaku, Tokyo, Japan), step of 0.01° 2*θ*, 20 °C). Le Bail fit refinement for powder diffraction pattern was performed using TOPAS, version 4 software package (Bruker, Billerica, MA, USA). Thermal gravimetric analysis (TGA) was performed on a TA-5500 machine (TA Instruments, New Castle, DE, USA) in a temperature range from 20 to 750 °C with a 25 mL/min argon flow at a heating rate of 1 °C/min. The NMR spectra were recorded on a Bruker Ascend-500 spectrometer (500 MHz for ^1^H) (Bruker, Billerica, MA, USA). Chemical shifts (δ) are reported in parts per million (ppm) and referenced to the resonances of the corresponding solvent used.

### 3.2. Synthesis of Hexanuclear Precursor

Synthesis of [Na(thd)] was carried out by a solution approach. Here, 500 mg of [NaOCH_3_] (9.3 mmol) was dissolved in 15 mL of methanol under argon, and then 2 mL of Hthd (1.76 g, 9.6 mmol) was added to the solution dropwise under the argon flow protection. The solution was initially light-yellow but turned colorless immediately after adding Hthd. The reaction was stirred at room temperature for an hour and methanol was removed under vacuum. White residue of [Na(thd)] was isolated and further dried under vacuum at 100 °C overnight. The yield was 1800 mg, ca. 94%. The ^1^H NMR spectrum of [Na(thd)] (^1^H NMR, CDCl_3_: 5.49 (s, C*H*, 1H), 1.00 (s, C*H*_3_, 18H)) and the DART mass spectrum of [Na(thd)] are shown in Appendix A.

Synthesis of hexanuclear complex [NaMn_2_(thd)_4_(OAc)]_2_ (**1**) was carried out by both solid-state and solution approaches through the stoichiometric reaction (Reaction 1). For the solid-state synthesis, a slight excess of [Na(thd)] 20 mg (0.097 mmol) and 5 mg (0.029 mmol) of [Mn(OAc)_2_] was sealed in an evacuated glass ampule. The ampule was placed in a furnace with a temperature gradient of 120 to 110 °C. After one week of heating, block-shaped light-yellow crystals of **1** were deposited in the cold zone of the container. The yield was 12 mg, ca. 92%. Microcrystalline powder of [NaMn_2_(thd)_4_(OAc)]_2_ (**1**) was obtained by dissolving 200 mg (0.97 mmol) of [Na(thd)] and 84 mg (0.49 mmol) of [Mn(OAc)_2_] in 20 mL of dry, deoxygenated ethanol under argon atmosphere. The solution turned yellow immediately and was stirred at room temperature for 1 h. The solvent was evaporated under vacuum and the reaction residue was dried at 80 °C for a few hours, and then redissolved in 30 mL of dry, oxygen-free hexanes and stirred for 30 min. After filtering off the white insoluble powder (sodium acetate), the solvent was removed under vacuum, and the yellow powder was dried at 80 °C in a sand bath overnight. The yield was 210 mg, ca. 94%.

### 3.3. Single-Crystal X-ray Investigation

The crystal of [NaMn_2_(thd)_4_(OAc)]_2_ (**1**) was immersed in cryo-oil, mounted on a glass fiber, and measured at a temperature of 100 (2) K. The X-ray diffraction data were collected on a Bruker SMART APEX CCD-based X-ray diffractometer system equipped with a Mo-target X-ray tube (λ = 0.71073 Å) operated at 1500 W power. The dataset reduction and integration were performed with the Bruker software package SAINT (version 8.37A) [39]. The data were corrected for absorption effects using the empirical methods as implemented in SADABS (version 2016/2) [40]. The structure was solved by SHELXT (version 2018/2) [41] and refined by full-matrix least-squares procedures using the Bruker SHELXTL (version 2019/2) [42] software package through the OLEX2 (version 1.5) graphical interface [43]. All non-hydrogen atoms were refined anisotropically. Hydrogen atoms were included in idealized positions for the structure factor calculations with U_iso_(H) = 1.2 U_eq_(C) and U_iso_(H) = 1.5 U_eq_(C) for the methyl groups. Crystallographic parameters and details of the data collection and structure refinement are listed in Table 5.

## 4. Conclusions

The first heterometallic single-source precursor for the sodium-manganese oxide cathode material Na_4_Mn_9_O_18_ is reported. The design of the hexanuclear molecule [NaMn_2_(thd)_4_(OAc)]_2_ (**1**) has been accomplished based on its parent analog, [LiMn_2_(thd)_5_], by replacing Na for Li and providing a higher coordination number by substituting one of the thd ligands with acetate to create the unique “dimer of trimers” assembly. The model structure of [LiMn_2_(thd)_5_], again, displays great potential for modifications. This molecule should be further explored for possible alterations. The design also represents another example of the mixed-ligand approach in application. Even the substitution of one out of five ligands was shown to bring about dramatic changes in the metal core. We have previously reported dinuclear, trinuclear, tetranucler, and pentanuclear molecules for Li(Na)-transition metal heterometallic precursors. The title molecule is the first example of a hexanuclear structure. Importantly, it contains four Mn positions that can be potentially substituted by other transition metals, leading to heteromultimetallic precursors for complex quaternary and even quinary Na-ion oxide cathode materials. 

## Figures and Tables

**Figure 1 molecules-28-07795-f001:**
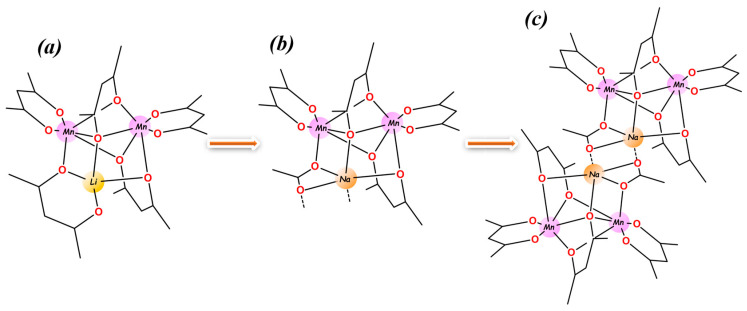
The design of [NaMn_2_(thd)_4_(OAc)]_2_ (**1**) molecule by replacing one thd ligand with acetate group from (**a**) the parent structure of [LiMn_2_(thd)_5_] to (**b**) [NaMn_2_(thd)_4_(OAc)] after substituting Li and thd ligand and (**c**) molecular structure of [NaMn_2_(thd)_4_(OAc)]_2_. *Tert*-butyl groups and hydrogen atoms are omitted for clarity.

**Figure 2 molecules-28-07795-f002:**
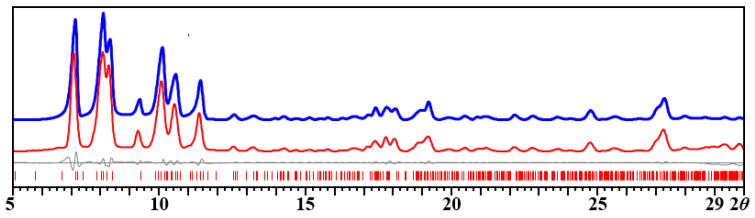
X-ray powder diffraction pattern of [NaMn_2_(thd)_4_(OAc)]_2_ (**1**) bulk product obtained from the solution reaction and the Le Bail fit. Blue and red curves represent experimental and calculated patterns, respectively. Gray is the difference curve with theoretical peak positions shown as red bars at the bottom.

**Figure 3 molecules-28-07795-f003:**
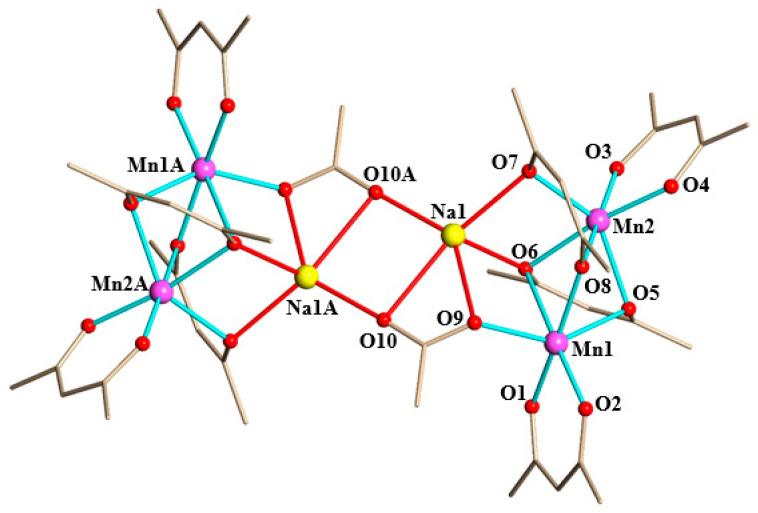
Molecular structure of [NaMn_2_(thd)_4_(OAc)]_2_ (**1**). *Tert*-butyl groups and hydrogen atoms are omitted for clarity. The Mn–O bonds are marked in blue and Na–O bonds in red. Full view of the structure with thermal ellipsoids, bond distances, and angles are included in the Appendix A.

**Figure 4 molecules-28-07795-f004:**
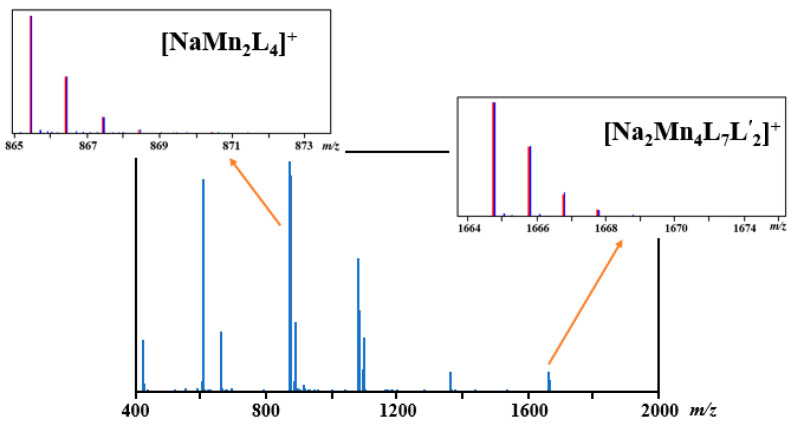
Positive DART-MS spectrum of [NaMn_2_(thd)_4_(OAc)]_2_ (**1**) measured at 275 °C. Isotope distribution patterns for heterometallic ions [M-L]^+^ and [NaMn_2_L_4_]^+^ (L = thd) are also shown in insets with experimental (blue bars) and theoretical (red bars) peaks.

**Figure 5 molecules-28-07795-f005:**
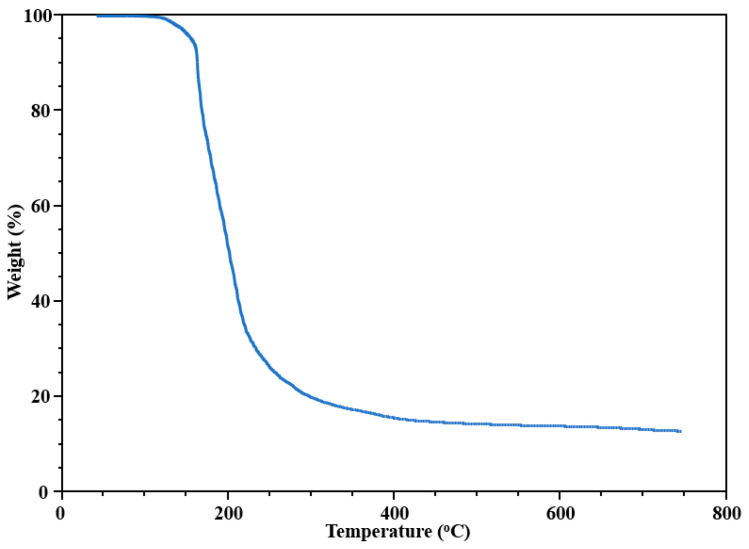
TGA plot of [NaMn_2_(thd)_4_(OAc)]_2_ (**1**).

**Figure 6 molecules-28-07795-f006:**
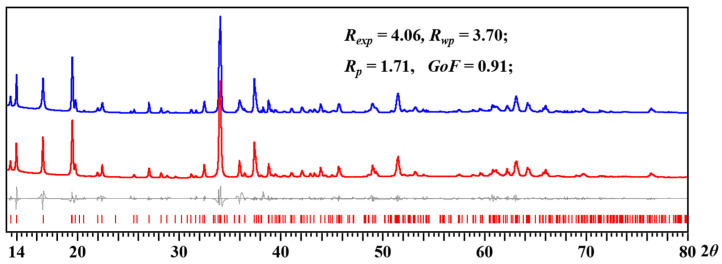
Powder X-ray diffraction pattern of Na_4_Mn_9_O_18_ oxide obtained by thermal decomposition of [NaMn_2_(thd)_4_(OAc)]_2_ (**1**) heterometallic precursor at 700 °C and the Le Bail fit. Blue and red curves represent experimental and calculated patterns, respectively. Gray is the difference curve with theoretical peak positions shown as red bars at the bottom.

**Table 1 molecules-28-07795-t001:** Comparison of the unit cell parameters for [NaMn_2_(thd)_4_(OAc)] (**1**) obtained from the single-crystal data and from the Le Bail fit.

	Single Crystal Data (−173 °C)	Le Bail Fit (20 °C)
Space group	*P*-1	*P*-1
*a* (Å)	12.466 (2)	12.9279 (14)
*b* (Å)	18.285 (4)	18.531 (2)
*c* (Å)	24.847 (5)	25.105 (3)
*α* (°)	81.650 (2)	81.533 (10)
*β* (°)	89.291 (2)	90.033 (12)
*γ* (°)	71.240 (2)	71.758 (10)
*V* (Å^3^)	5302.6 (18)	5642.7 (11)

**Table 2 molecules-28-07795-t002:** Comparison of the key distances and angles in the structures of [NaMn_2_(thd)_4_(OAc)]_2_ (**1**) and [LiMn_2_(thd)_5_]; (A = Na, Li).

Average Distances (Å)	[NaMn_2_(thd)_4_(OAc)]_2_	[LiMn_2_(thd)_5_]
Mn1–O *	2.069 (3)	2.069 (3)
Mn1–O **	2.290 (3)	2.253 (3)
Mn2–O *	2.061 (3)	2.060 (3)
Mn2–O **	2.195 (3)	2.209 (3)
A–O	2.367 (3)	1.966 (7)
Mn1···Mn2	3.105 (3)	3.161 (3)
Mn1···A	3.424 (3)	2.884 (3)
Mn2···A	3.405 (2)	2.939 (3)
**Angles (°)**		
Mn1–A–Mn2	54.08 (3)	65.76 (2)
A–Mn1–Mn2	62.56 (3)	57.95 (2)
A–Mn2–Mn1	63.27 (3)	56.29 (2)

* chelating oxygen; ** bridging oxygen.

**Table 3 molecules-28-07795-t003:** Assignment of ions detected in a positive-ion DART mass spectrum of [NaMn_2_(thd)_4_(OAc)]_2_ (**1**) (L = thd, L’ = OAc). All ions having relative intensity higher than 3% are shown in the table.

Fragments	Calc. *m*/*z*	Exp. *m*/*z*	Δ	Relative Int. (*%*)
[Na_2_Mn_4_L_7_L’_2_]^+^	1665.728	1665.738	0.010	9.0
[Na_2_Mn_3_L_6_L’]^+^	1368.638	1368.647	0.009	8.9
[Mn_3_L_5_(H_2_O)]^+^	1098.517	1098.550	0.033	24.1
[Mn_3_L_5_]^+^	1080.507	1080.521	0.014	58.4
[NaMn_2_L_4_(H_2_O)]^+^	883.430	883.850	0.010	30.4
[NaMn_2_L_4_]^+^	865.420	865.424	0.004	100
[Mn_2_L_3_]^+^	659.292	659.304	0.012	26.6
[H_2_MnL_3_]^+^	606.369	606.377	0.012	92.5
[HMnL_2_]^+^	422.223	422.239	0.016	22.5

**Table 4 molecules-28-07795-t004:** Comparison of the unit cell parameters for Na_4_Mn_9_O_18_ product obtained by thermal decomposition of single-source precursor **1** with the data from the PDF-2 database (01-080-7228).

	Le Bail Fit (20 °C)	PDF-2 Database [38]
Space group	*Pbam*	*Pbam*
*a* (Å)	9.096 (2)	9.084 (4)
*b* (Å)	26.371 (10)	26.311 (10)
*c* (Å)	2.826 (3)	2.8223 (11)
*V* (Å^3^)	677.9 (7)	674.6 (8)

**Table 5 molecules-28-07795-t005:** Crystal data and structure refinement parameters for [NaMn_2_(thd)_4_(OAc)]_2_ (**1**).

Compound	[NaMn_2_(thd)_4_(OAc)]_2_
CCDC	2,301,967
Empirical formula	C_92_H_158_Mn_4_Na_2_O_20_
Formula weight	1849.91
Temperature (K)	100 (2)
Wavelength (Ǻ)	0.71073
Crystal system	Triclinic
Space group	*P*-1
*a* (Å)	12.466 (2)
*b* (Å)	18.285 (3)
*c* (Å)	24.847 (4)
*α* (°)	81.650 (2)
*β* (°)	89.291 (3)
*γ* (°)	71.240 (2)
*V* (Å^3^)	5302.6 (16)
*Z*	2
*ρ*_calcd_ (g·cm^−3^)	1.159
*μ* (mm^−1^)	0.533
*F* (000)	1984
Crystal size (mm^3^)	0.15 × 0.06 × 0.03
*θ* range for data collection (°)	1.548–28.069
Reflections collected	45,149
Independent reflections	23,476
Transmission factors (min/max)	0.6255/0.7457
Completeness to full *θ* (%)	98.4
Data/restraints/params.	23,476/729/1378
*R*1, ^a^ *wR*2 ^b^ (*I* > 2*σ*(*I*))	0.0510/0.1240
*R*1, ^a^ *wR*2 ^b^ (all data)	0.0940/0.1432
Quality-of-fit ^c^	0.975

^a^*R*1 = Σ||*F*_o_| − |*F*_c_||/Σ|*F*_o_|. ^b^
*wR*2 = [Σ[*w*(*F*_o_^2^ − *F*_c_^2^)^2^]/Σ[*w*(*F*_o_^2^)^2^]]. ^c^ Quality-of-fit = [Σ[*w*(*F*_o_^2^ − *F*_c_^2^)^2^]/(*N*_obs_ − *N*_params_)]^½^, based on all data.

## Data Availability

Data in this study are available upon request.

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
