# Peer review of "Synthesis, Structure, and Characterizations of a Volatile/Soluble Heterometallic Hexanuclear Precursor [NaMn2(thd)4(OAc)]2"

_molecules, 2023, doi:10.3390/molecules28237795_

Round 1

Reviewer 1 Report

Comments and Suggestions for Authors

1) Chapter 2.2. - in my opinion, the description of TGA should be combined within the appropriate chapter. In addition, the description contains the characteristic “sharp color change”. It is not specified which colors are meant. In this part of the article, the description of thermolysis is very vague.

2) Chapter 2.2 – “Strongly coordinating solvents like DMSO and H2O appeared to break up the heterometallic structure.” - How was this phenomenon proven?

3) It is unclear what the authors mean by repeatedly mentioning in the text that the complex is “highly volatile”. There is no evidence or characteristics given that would indicate this, other than the fact of solubility in various solvents.

4) Chapter 2.3 with a description of TGA should be renamed 2.5. Description of TGA - temperature intervals, mass loss values, comparison with theoretical ones should be given. It is unclear what the authors mean by “small mass loss due to volatility.”

5) Chapter 2.3. - The authors give the geometry of metal polyhedra. It is advisable to characterize the distortion of polyhedra using software methods, for example, SHAPE 2.1 software [Llunell, M.; Casanova, D.; Cirera, J.; Alemany, P.; Alvarez, S. SHAPE v.2.1; Program for the Stereochemical Analysis of Molecular Fragments by Means of Continuous Shape Measures and Associated Tools; Universitat de Barcelona: Barcelona, Spain, 2013.] 6) In my opinion, it is not entirely correct to provide an analysis of the geometry of the sodium polyhedron since the nature of the sodium-oxygen bonds is ionic and we can rather talk about the nearest environment of donor atoms. 7) The authors do not provide details of the refinement of structural parameters based on Powder X-ray diffraction data (for example, GOF, information about which parameters varied during refinement), but only general phrases about the measurements. In my opinion, information about refinement of cell parameters using PXRD is unnecessary. 8) Chapter 3 – throughout the entire article acetate was designated OAc; there is no reason to introduce a new designation in the experimental part. 9) Chapter 3.2. – the authors provide the formula for the previously isolated precursor [Na(thd)]. How was this particular composition of the compound proven? Data must be provided. 10) Conclusion. "Importantly, it contains four Mn positions that can be substituted by other transition metals leading to heteromultimetallic precursors for complex quaternary and even quinary Na-ion oxide cathode materials." In my opinion, the conclusion is incorrect since the article does not contain any information about such a possibility. The phrase should be formulated with an emphasis on the fact that this is the authors' assumption. In addition, this phrase should be removed from the abstract (the last sentence in the abstract).

Reviewer 2 Report

Comments and Suggestions for Authors

Prof. Dikarev and co-workers have successfully synthesis of a unique heterometallic hexanuclear complex. The structure and properties of the complex have been investigated perfectly. The reported complex should be the good precursor of the novel multinuclear metallic materials. This is a thorough study and a well-written paper. It would rank in the top 5% of work I have reviewed for this journal. I have only one request for the author. IR spectrum of the complex is given – but not discussed. The authors should do that.

Author Response

Dear Reviewer, 

From our previous works on heterometallic diketonates, we came to the conclusion that it is nearly impossible to analyze their IR spectra without sophisticated calculations. In the course of our previous work, we have only pointed out to the C–H stretching frequencies that occur at around 3000 cm-1 (Inorg. Chem. 2008, 47, 10046; J. Organomet. Chem. 2009, 694, 2956). For complex 1 we can see the aromatic C–H stretches in addition to C–H (methyl) bands for methyl groups of thd and acetates at around 3000 cm-1. The stretching frequencies in the range of 1600 to 1000 cm-1 correspond to C–O (aromatic), C–C (aromatic), and C–R bonds and are hard to be specifically assigned. As an example, even in the simple manganese diketonate [Mn(acac)3], there are 8 stretches between 1600 and 1000 cm-1 (Spectrochim. Acta, 1961, 17, 248) In the case of 1, we have two Mn ions in different coordination environments and carboxylate ligands on top of this. Apparently, one can see a severe overlap in this range. The frequencies lower than 1000 cm-1 primarily correspond to the Mn–O and Na–O bonds (among others). Again, those are not easy to be unambiguously assigned.

Round 2

Reviewer 1 Report

Comments and Suggestions for Authors

The manuscript can be published in the present form.